# Insight into the Optimization of Implementation Time in Cob Construction: Field Test and Compressive Strength Versus Drying Kinetics

**Karim Touati** [1,2,*], **Baraa Al Sahmarany** [2], **Malo Le Guern** [2], **Yassine El Mendili** [3], **François Streiff** [4] **and Steve Goodhew** [5]

1   EPF Ecole d'Ingénieurs, 21 Boulevard Berthelot, 34000 Montpellier, France
2   ComUE Normandie Université, Builders Ecole d'Ingénieurs, 1 Rue Pierre et Marie Curie, 14610 Epron, France; baraa.alsahmarany96@gmail.com (B.A.S.); malo.leguern@builders-ingenieurs.fr (M.L.G.)
3   Institut de Recherche en Constructibilité IRC, Ecole Spéciale des Travaux Publics, 28 Avenue du Président Wilson, 94234 Cachan, France; yelmendili@estp-paris.eu
4   Parc Naturel Régional des Marais du Cotentin et du Bessin, 50500 Carentan les Marais, France; fstreiff@parc-cotentin-bessin.fr
5   School of Art, Design and Architecture, University of Plymouth, Plymouth PL4 8AA, UK; s.goodhew@plymouth.ac.uk
*   Correspondence: karim.touati@epf.fr; Tel.: +33-4-99-65-99-55

**Abstract:** Mastering construction times is of paramount importance in making vernacular earth construction techniques attractive to modern clients. The work presented here is a contribution towards the optimization of the construction time of cob buildings. Therefore, this paper follows the evolution of a cob's mechanical properties during its drying process in the case of a double-walling CobBauge system. Laboratory tests and in situ measurements were performed, and further results were described. Volumetric water content sensors were immersed in the walls of a CobBauge prototype building during its construction. The evolution of the cob layer's compressive strength and Clegg Impact Value (CIV) as a function of its water content has been experimentally studied and discussed. These studies showed that compressive strength and CIV are correlated with water content, and both properties decrease exponentially with time. In this study, a new tool to evaluate cob's mechanical performances in situ has been proposed, Clegg Impact Soil Tester. This was linked to compressive strength, and a linear relationship between these two properties was found. Finally, appropriate values of compressive strength and CIV to satisfy before formwork stripping and re-lifting were proposed. For this study's conditions, these values are reached after approximately 27 days.

**Keywords:** implementation time; cob; water content; compressive strength; Clegg Impact Value





## 1. Introduction

Climate change has been observed worldwide over recent decades. This is in major part due to greenhouse gas emissions (GHG). In France, the sector of the manufacturing and construction industry, in particular, related to the use and construction of residential/tertiary buildings contributed to 152.7 $MtCO_2eq$ in 2021, representing 36.5% of total GHG emissions [1]. Thus, construction is one of the major target sectors that should be focused on to reduce carbon footprints. This can be achieved by specifying less processed materials, using locally sourced natural materials, and low environmental impact processes. In the context of global climate change, the development of earthen construction is a real alternative to reduce the $CO_2$ emissions from the construction sector. However, this age-old material must be able to demonstrate good mechanical resistance and adequate water resistance whilst complying with conditions imposed by the current building regulations.

Earthen construction is gaining popularity as a potential means of establishing local value chains with minimal environmental impact. However, the growth of this historically existing building technology, concentrated in a number of locations of developed countries is still limited due to the high cost, labour intensity, and construction periods because of the material drying time. One of the most popular earthen construction techniques in the Northwest of Europe is cob. Natural fibers, water, and silty-clayey soils are typically used to make cob. Water is added to the mixture to cause it to transform into a plastic state, which enables the efficient production of reasonably thick load-bearing building walls. Cob has gained less interest since the beginning of the 20th century in favor of industrial materials thought to be more effective and contemporary with a high degree of standardization. When compared to modern construction methods, cob has actually numerous perceived drawbacks, including low insulating capabilities, long construction durations, high labor demands, etc. [2]. Actually, buildings using cob (in its original conception) do not adhere to global thermal construction rules. To overcome this issue and allow building designers to use cob as a walling material, some aspects of the properties of the finished walling need to be improved in order to enhance cob housings' thermal performances. Alongside this need, the implementation of new methods is being examined to reduce construction times and associated costs.

Cob- and earthen-based materials, in general, have seen a rise in attention since the 1980s due to climatic concerns, especially in France and the United Kingdom [3–11]. This is partly because of its many environmental benefits, namely its durability, minimal environmental effect, and occupational thermal comfort [12,13]. Understanding and improving cob's geotechnical, thermal, and mechanical properties has been the subject of several studies [7,12,14,15].

Accordingly, the CobBauge EU Interreg project is concentrated on creating, implementing, and testing a novel low-carbon technology employing regional soils and plant fibers. This earth fiber-based technique attempts to create a hybrid walling system by combining a load-bearing (cob) and an insulating (light earth) layer, as shown in Figure 1. In order to create 100% earth-based walling that ensures structural resistance and thermal insulation, cob and light earth are naturally blended. In the first phase of the project, many formulations have been examined and evaluated to identify the best earth/fiber mixtures that would enable the construction of the building with a double walling system that complies with French and United Kingdom standards. In this first phase, studies focused on the geotechnical properties of soils, followed by the mechanical and thermal properties of soil–fiber composites. Two prototype buildings have been built in France and the United Kingdom as part of the project's second phase in order to evaluate how the CobBauge hybrid walling affects the behavior of those buildings on the site.

When compared to other modern construction techniques, a building with a cob represents an issue possibly limiting its large-scale rollout. This concerns construction times. With cob, construction times are usually long and sometimes weather dependent. These long construction times are mainly due to the leakage of knowledge regarding the drying kinetics of the mixtures. Thus, the first aim of this present work is to understand the cob's drying process in situ and propose a methodology to optimize the walling's construction time. This objective can be achieved by studying and understanding the impact of water content on the mechanical performances of cob. The second aim of this work is to propose a field test allowing the knowledge of in situ performances (drying kinetics and compressive strength). The results will make it possible to evaluate cob drying and compressive strength progress in situ. Since a double walling system is implemented within formwork, this contribution can be useful in the optimization of form-stripping times, particularly, influencing cob construction times, in general.

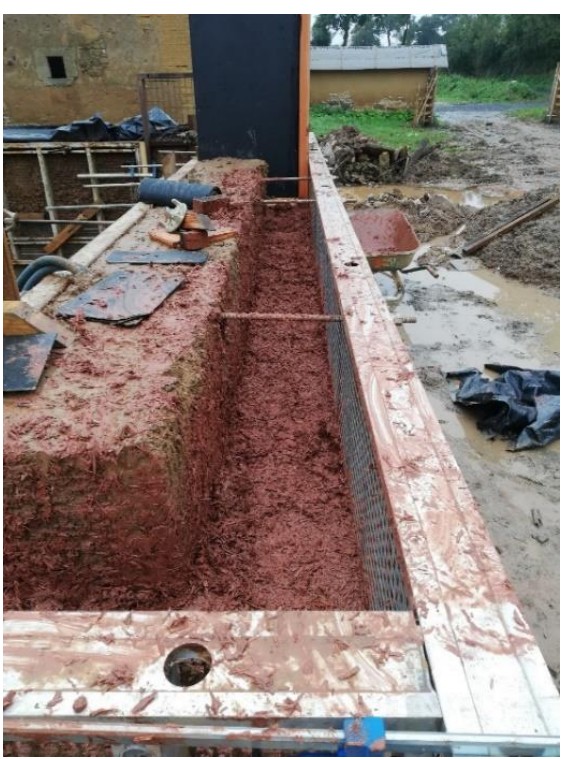

**Figure 1.** In situ implementation of a CobBauge double walling system.

## 2. Materials and Methods

### 2.1. Soils Particle Size and Geotechnical Characterization

Three soils were used in the construction of a prototype building in France, designed to allow the research team to assess the construction process performances and energy efficiency of the CobBauge technique. The cob layer (subject of this present paper) is constituted of two different soils. The soils used in this study were collected in Lieusaint quarry (Société des Sablières du Cotentin, SABCO Normandy). These soils are locally sourced from a geographical area that is associated with existing cob buildings. Their geotechnical characterizations were performed. The clay activity was evaluated using the methylene blue value test according to standard NF P94-068 [16] and Atterberg's limit according to standard NF EN ISO 17892-12 [17]. Retrieved soil properties constituting cob mixture and classification can be found in Table 1.

**Table 1.** Atterberg limits, methylene blue values, and soil classification.

| Soil | Liquid Limit [%] | Plasticity Index [%] | Methylene Blue Value [g/100 g] | USCS Classification |
|------|------------------|----------------------|-------------------------------|---------------------|
| Soil 1 | 22.8 | 2.3 | 1.35 | Low plasticity silt (ML) |
| Soil 2 | 28.5 | 4.2 | 2.31 | Silty sand with gravel (SM) |

The mineralogical composition of soil 1 reveals the presence of the following major phases: quartz (54.8%), muscovite (26.2%), montmorillonite (6.9%), and albite (4.2%), with small occurrences of illite, kaolinite, goethite, rutile, and huntite (see Table 2) [18]. Subsequently, soil 1 is typical of silty soil. It is composed of quartz grains and silicates (feldspars, micas, serpentines, and smectites). Silt particles are intermediate sand and clay in size and have similar properties.

**Table 2.** Mineralogical composition of soil 1 with refined values of unit cell volume and average diameter. One standard deviation is indicated in parenthesis on the last digit.

| Phases | V (%) | $\langle D \rangle$ (nm) |
|---|---|---|
| Quartz $SiO_2$ | 54.8 (5) | 492 (10) |
| Muscovite $KAl_2(AlSi_3O_{10})(F,OH)_2$ | 26.2 (5) | 35 (5) |
| Montmorillonite $(Na,Ca)_{0.3}(Al,Mg)_2Si_4O_{10}(OH)_2$ | 6.9 (2) | 111 (6) |
| Albite $NaAlSiO_3$ | 4.2 (2) | 43 (5) |
| Kaolinite $Al_2Si_2O_5(OH)_4$ | 2.1 (3) | 78 (5) |
| Goethite $\alpha$-FeO(OH) | 2.0 (3) | 21 (1) |
| Rutile $TiO_2$ | 1.6 (3) | 92 (5) |
| Illite $(K,H_3O)(Al,Mg,Fe)_2(Si,Al)_4O_{10}[(OH)_2,(H_2O)]$ | 1.1 (2) | 100 (5) |
| Huntite $Mg_3Ca(CO_3)_4$ | 1.1 (2) | 123 (5) |

Soil 2 is made of natural quartz (99% silica) [18].

### 2.2. Flax Straw

The selection of flax straw was based on the results of an earlier study [14], which established that flax straw offered earth–fiber mixes optimum compressive strength when dried while retaining good workability during mixing. Flax straw incorporated into the soil represents theoretically a proportion of 2.5% of the mixed dry mass. In laboratory, the flax straw is cut to a length of $7 \pm 1$ cm. On site, flax straw was introduced into the mix in its raw state. This latter presents an absolute density equal to $1266 \pm 55$ kg·m$^{-3}$ and an absorption coefficient at 24 h equal to $350 \pm 11\%$.

### 2.3. Sample Preparation

To study cob's compressive strength as a function of water content, 24 specimens were prepared at the same time with same proportion of soil, straw, and water. To be representative of what can be encountered in situ after cob implementation, 8 different water contents are considered: 19%, 17%, 15%, 13%, 11%, 9%, 5%, and 0%. Three samples are made for each water content. Cylindrical molds with the following dimensions were used: Ø110 mm × H220 mm. Before starting the production of the specimens, the mold's inner face was oiled. Afterwards, the mixture was compacted in several layers (a new layer is added when the previous one is considered completely compact) with a wooden tamper with dimensions equal to 30 mm × 30 mm × 410 mm (see Figure 2). Then, 24 h after filling the molds, the samples were turned upside down in order to ensure a good distribution of the water in them. In addition, the molds are slightly opened to accelerate the drying of the mixtures. Then, these samples are demolded after two or three days and left to dry until reaching their required water content.

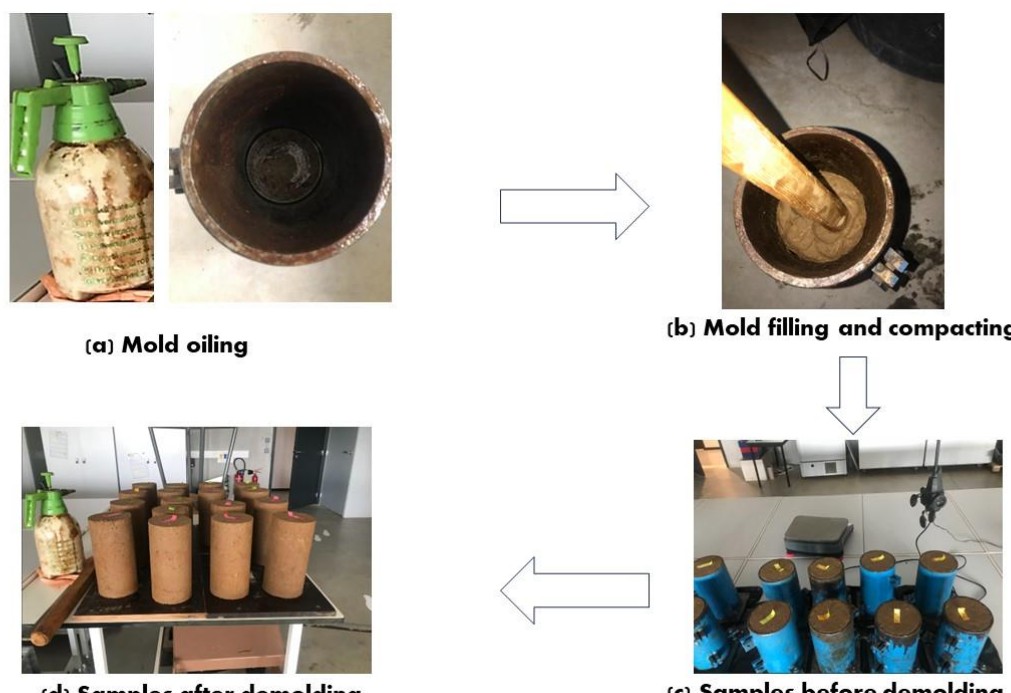

**Figure 2.** Preparation of cob samples at different water contents for the compressive strength study.

*2.4. Water Content Control and Bulk Density*

To better control the drying process, the samples were placed in the laboratory at a temperature of 21 ± 2 °C and a relative humidity of 50 ± 2%. When the required moisture content was reached, the samples were covered for 48 h to homogenize the water content within them, and their bulk density was measured before performing the compressive strength test (see Figure 3). The bulk density was determined by following the standard NF X31-501 [19]. Successive measurements of the sample's weight are made until reaching and stabilizing at the corresponding required water content. Samples with following water content have been prepared: 19, 17, 15, 13, 11, 9, 5, and 0%. These percentages have been chosen to approach values encountered on site during the cob's drying course.

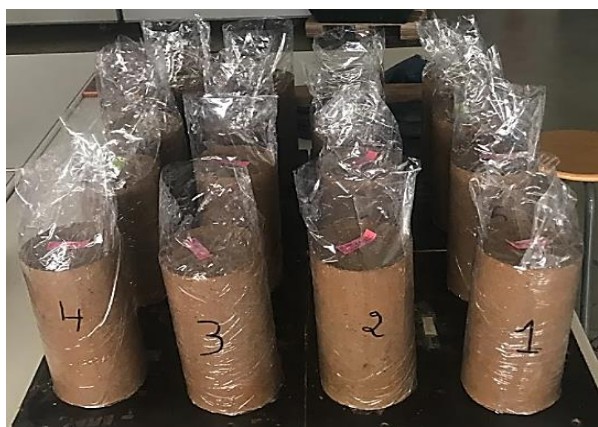

**Figure 3.** Samples coverage to homogenize water content (the samples are numbered from 1 to 24).

*2.5. Compressive Strength*

To maintain a measure of consistency between the samples, enabling the research team to relate the structural performance of the samples to the needs of the walling system, compressive strength measurements were undertaken on samples with different water contents. Compressive strength tests are performed on the prepared cylindrical samples

(∅110 mm × H220 mm) in accordance with the NF EN 13286-41 standard [20]. An IGM press with a load capacity of 250 kN was used. The tests are performed with an imposed loading rate of 0.40 kN/s. The deformation of the specimens is measured with a vertical displacement sensor in contact with the lower plate of the press (see Figure 4).

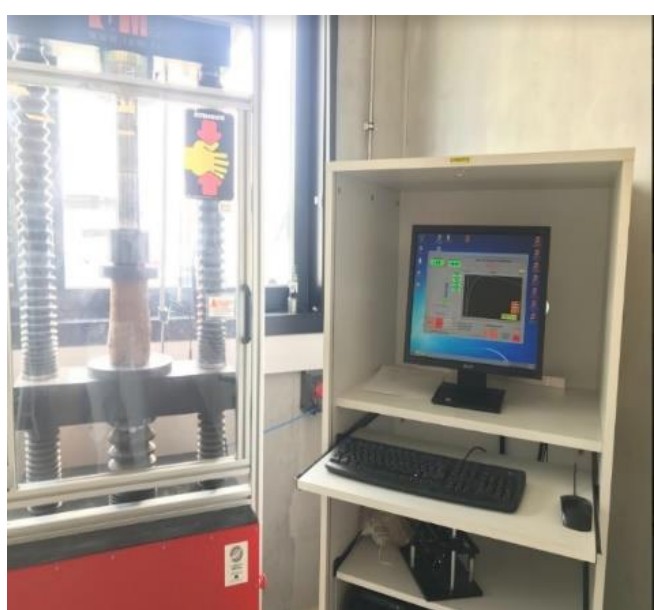

**Figure 4.** Compressive strength test on cob samples at different water content under an IGM press.

*2.6. Clegg Impact Value*

The Clegg Impact Soil Tester (CIST) was proposed by Dr. Bade Clegg in the 1970s as an alternative to the CBR test. The output of the CIST device is called the Clegg Impact Value (CIV). In this present study, this value will be linked to the compressive strength in order to propose a site control test. To study Clegg Impact Value, 10 specimens were prepared. To be representative of what can be encountered in situ, 5 different water contents have been considered: 19%, 15%, 11%, 5%, and 0%. For mold availability considerations, two samples are made for each water content. CBR molds with the following dimensions were used: ∅152 mm × H117 mm. Preparation method is quite similar to the one described in the precedent paragraph (see Figure 5).

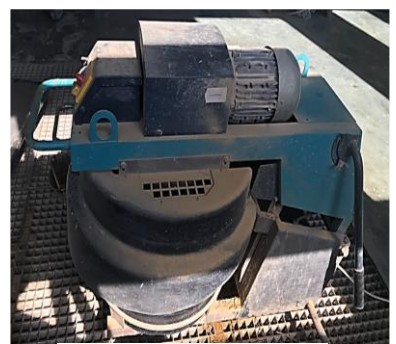
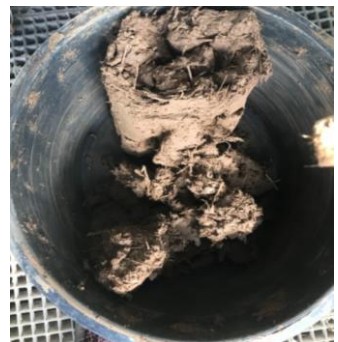
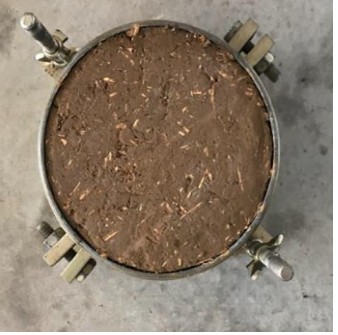

Mixer      Material mixing      CBR molds filled

**Figure 5.** Preparation of cob samples at different water contents for the Clegg Impact Value measurement.

The mobility of the CIST makes it possible to perform in situ tests relatively quickly. Initially, the CIST device was developed for non-cohesive backfill materials. Clegg impact soil tester is implemented in accordance with ASTM D5874-02. This method using a hammer of 4.5 kg is suitable for soils or aggregates with particle sizes less than 37.5 mm.

The soils used in this present study have particle sizes less than 20 mm (see Figure 6). The impact height is equal to 0.45 m. The hammer diameter is 0.05 m. The CIV value is obtained using the knowledge of gravitational acceleration (g = 9.81 m·s$^{-2}$) and the deceleration measured during the hammer drop (a in m·s$^{-2}$) is expressed as the following equation [21]:

$$CIV = \frac{a}{10 \cdot g} \qquad (1)$$

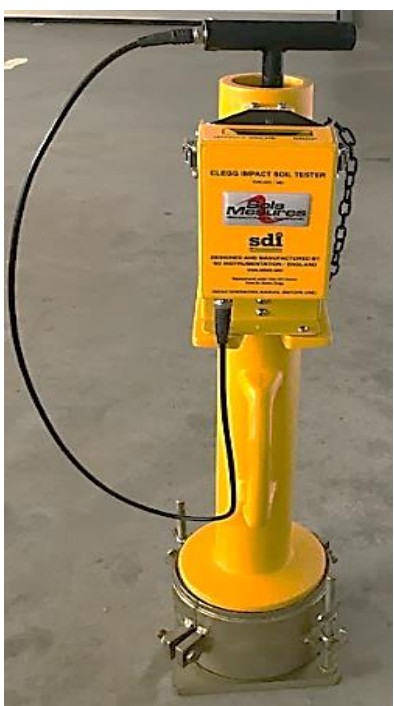

**Figure 6.** Clegg Impact Soil Tester on cob samples at different water contents.

The Clegg Impact Value can be used to calculate quantities such as the modulus of elasticity or the CBR (California Bearing Ratio) value using correlations. In this study, CIV is linked to cob's compressive strength. This can represent an easy way for craftsmen to control cob's resistance in situ. This can be interesting to know if a cob lift is sufficiently dry and resistant to receive a new lift on its top.

*2.7. In Situ Implementation and Monitoring*

The main focus of this research is the drying rates of CobBauge dual walls. It is, therefore, important to undertake moisture measurements of representative samples. In this present work, Campbell scientific CS655 sensors, based on reflectometry principle, have been used to locally measure Volumetric Water Content (VWC). This type of sensor (with an accuracy of ±3%) has shown its efficiency when measuring VWC in soil materials [22–24].

The theory behind the CS655 water content reflectometer is based on the speed at which an electromagnetic wave travels through the sensor's two rods. The latter depends on the material's dielectric permittivity surrounding the two rods. Dielectric permittivity is then converted to volumetric water content using the Topp equation [25]. For the sake of greater contact between mixes and rods, sensors were positioned horizontally in cob at the same heights and depths. Probes were positioned parallel to the west wall surfaces at two different heights, 25 and 50 cm, from the lift basis. Volumetric water content considered in this present work is an average of these two considered heights. The thickness of cob in this west wall is equal to 25 cm. Additionally, information gathered via CS 655 probes was logged in the CR1000X data logger every 15 min during the course of more than a year (from May 2020 to November 2021). The following weather conditions are presented in

Figure 7: temperature, relative humidity, wind speed and direction, radiation, and rainfall recovered every 15 min. Instrumentation implemented on site is shown in Figure 8.

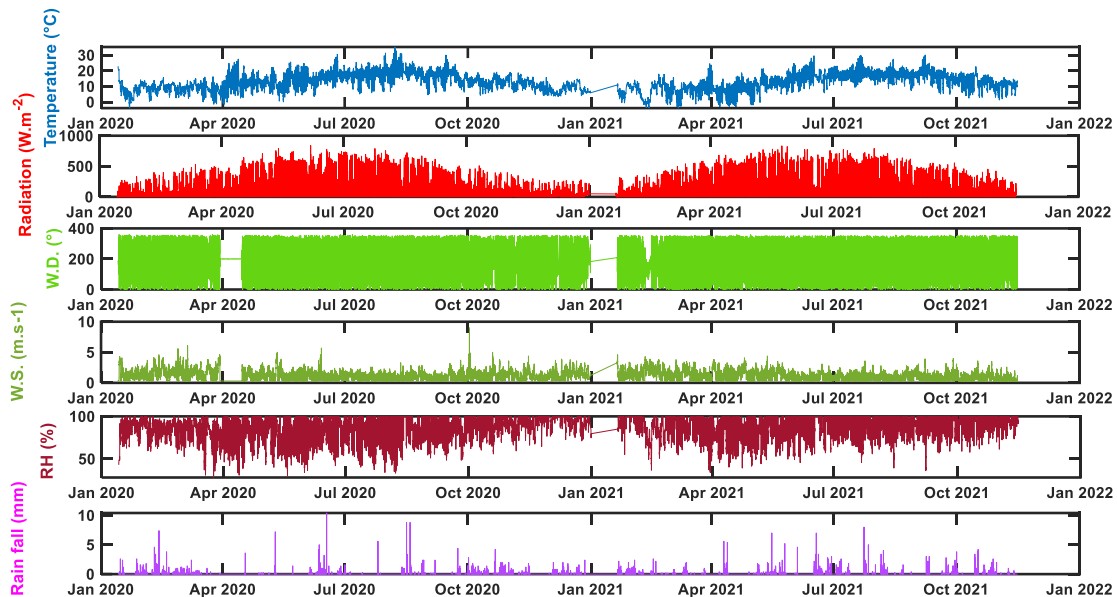

**Figure 7.** Weather conditions (temperature, RH, W.D., W.S., radiation, rainfall) recovered via a WS-GP1 weather station near the prototype building.

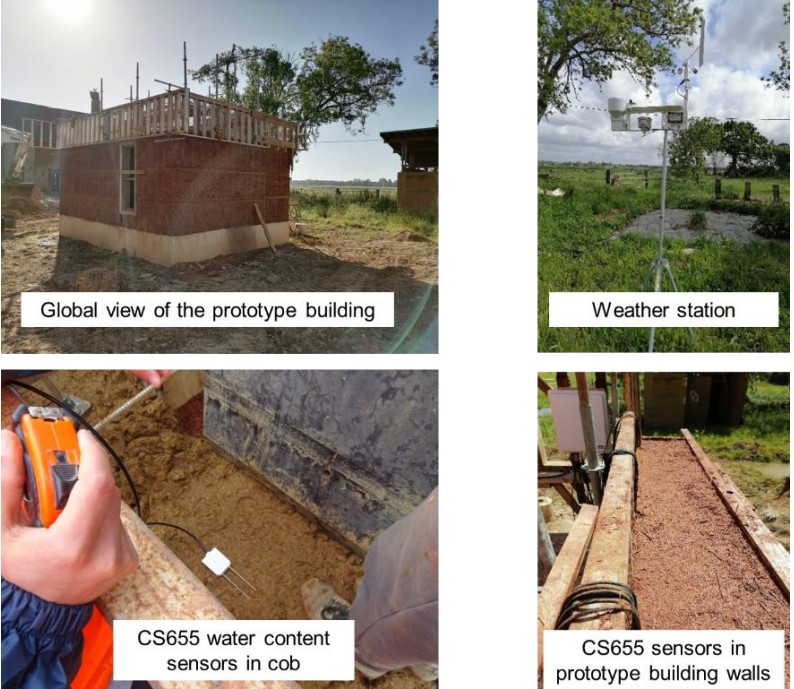

**Figure 8.** Instrumentation (weather station and volumetric water content sensors) implemented in situ.

## 3. Results and Discussion

### 3.1. Cob's Compressive Strength and Density as Function of Water Content

Figure 9 shows the different types of failure (deformation) that occurred in the samples after the compressive loading. When the water content is above 11%, there is crushing of the sample, and plastic behavior is predominant. However, when the water content is equal to or less than 11%, the sample breaks, and cracks are present throughout the sample.

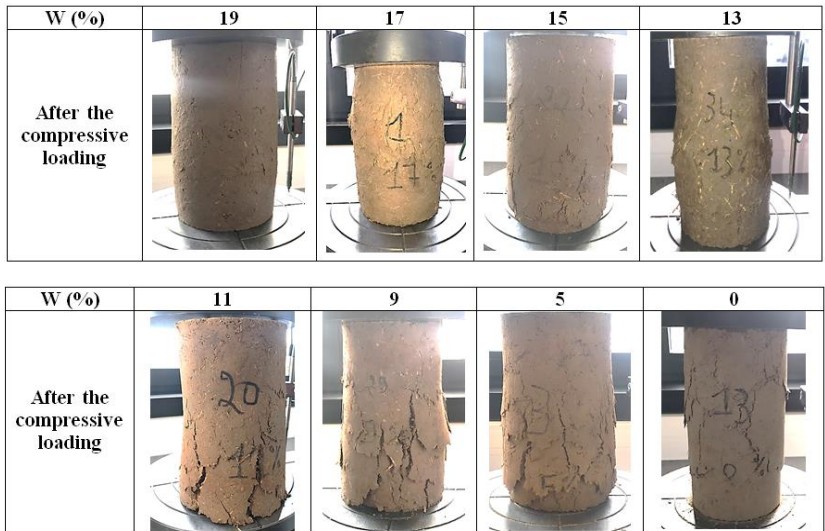

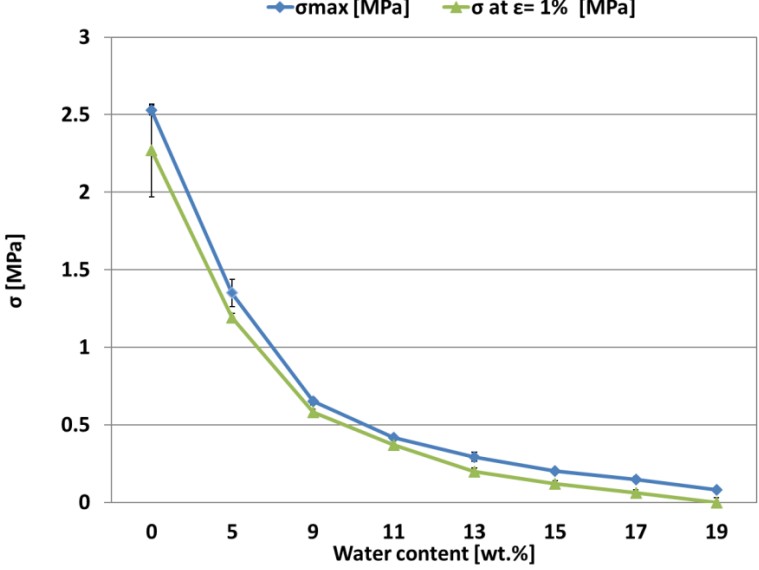

**Figure 9.** Cob's samples facies after compressive loading at different water content.

The evolution of the cob's compressive strength and density as a function of water content is reported in Table 3 and Figure 10. On this latter, maximum compressive strength and one at 1% deformation are presented. A deformation of 1% is considered in this present work for considerations regarding the ease of straightening wall surfaces before the application of plasters or renders.

**Table 3.** Evolution of compressive strength and Clegg Impact Value as function of water contents usually encountered in situ.

| WC [wt.%] | VWC [m³·m⁻³] | Density [kg·m⁻³] | $\sigma_{\varepsilon = 1\%}$ [MPa] | $\sigma_{max}$ [MPa] | CIV [-] |
|---|---|---|---|---|---|
| 19 | 0.3672738 | 1933.02 ± 15.65 | 0.04 ± 0.03 | 0.08 ± 0.01 | 04.0 ± 0.0 |
| 17 | 0.3315272 | 1950.16 ± 09.67 | 0.06 ± 0.02 | 0.15 ± 0.01 | - |
| 15 | 0.2822145 | 1881.43 ± 35.80 | 0.12 ± 0.02 | 0.20 ± 0.00 | 08.0 ± 1.4 |
| 13 | 0.2425423 | 1865.71 ± 05.37 | 0.20 ± 0.02 | 0.29 ± 0.03 | - |
| 11 | 0.2002693 | 1820.63 ± 05.64 | 0.37 ± 0.03 | 0.42 ± 0.01 | 12.5 ± 0.7 |
| 9 | 0.1625859 | 1806.51 ± 02.25 | 0.58 ± 0.02 | 0.65 ± 0.02 | - |
| 5 | 0.0868015 | 1736.03 ± 10.24 | 1.19 ± 0.03 | 1.35 ± 0.03 | 20.0 ± 1.4 |
| 0 | 0 | 1656.98 ± 08.11 | 2.27 ± 0.30 | 2.60 ± 0.03 | 34.0 ± 1.4 |

**Figure 10.** Evolution of cob's compressive strength as function of water content.

The clay content is critical in earth building because it maintains the larger particles connected. However, soils containing more than 30% clay have very high shrinkage/swelling ratios, which, along with their proclivity to absorb moisture, can result in large fissures in the finished cob product and thus impact its mechanical performance. The soil 1 is composed of quartz (54.8%), muscovite (26.2%), montmorillonite (6.9%), and albite (4.2%), with minor traces of kaolinite, goethite, rutile, illite, and huntite. Some clays have the ability to widen the interfoliar spaces between their leaves. The incorporation of hydrated cations (Na, Ca, etc.) provides it with this property, allowing it to compensate for chronic charge shortages (Andrade et al., 2011). The phenomenon disappears if the clay charge is too high (e.g., micas or muskovite in our sample: total clay charge of $-1$ entirely counterbalanced by the dehydrated cations (e.g., pyrophyllite, talc: total clay charge of 0, no interfoliar cation). With a charge ranging from 0.3 to 0.8, the smectites subclass is among the expandable species. The crystalline structure might expand due to the water injected via the hydrated cations [26]. The swelling increased as a result of the high humidity. Montmorillonite is the only expandable species identified in our soil 1, at a rate of 6.9%. The amount of muscovite, albite, kaolinite, and illite in our soil will impact its shrinkage characteristics. These crystals have few water molecules between their layers due to their tiny interfoliar space [26]. As a result, they have negligible intercrystalline swelling when immersed in water [27]. As a result, when dried, these four species shrink significantly less than smectite clays like montmorillonite [28]. Considering the number of smectites, the cob layer shrinkage properties are mainly affected by the quantity of muscovite, albite, kaolinite, and illite. These structures have few water molecules between their layers, and they exhibit negligible intercrystalline swelling and shrink substantially less than smectite clays like montmorillonite.

To explain the influence of the drying processes, the phase composition of soil 1 after drying was determined via XRD (Table 4). The corresponding XRD pattern is shown in Figure 11. The XRD analysis shows the disappearance of huntite combined with the decrease in the montmorillonite content in favor of the formation of carbonated calcium hemicarboaluminate (see Tables 2 and 3). Montmorillonite is the only expandable species found in the structural cob. The presence of kaolinite, muscovite, and illite, as well as the reduction in montmorillonite, will decrease the shrinking behavior. Indeed, these crystals have a weak intercrystalline swelling behavior and contain a minor amount of water [26]. The formation of carbonated calcium hemicarboaluminate leads to the enhancement of compressive strength (see Figure 10).

**Table 4.** Mineralogical composition of soil 1 after the drying process.

| Phases | V (%) | $\langle D \rangle$ (nm) |
|---|---|---|
| Quartz $SiO_2$ | 61.2 (5) | 492 (10) |
| Muscovite $KAl_2(AlSi_3O_{10})(F,OH)_2$ | 26.0 (5) | 34 (5) |
| Montmorillonite $(Na,Ca)_{0.3}(Al,Mg)_2Si_4O_{10}(OH)_2$ | 0.9 (2) | 67 (5) |
| Albite $NaAlSiO_3$ | 1.7 (2) | 46 (5) |
| Kaolinite $Al_2Si_2O_5(OH)_4$ | 1.5 (3) | 78 (5) |
| Illite $(K,H_3O)(Al,Mg,Fe)_2(Si,Al)_4O_{10}[(OH)_2,(H_2O)]$ | 2.1 (2) | 100 (5) |
| Rutile $TiO_2$ | 3.4 (3) | 92 (5) |
| Carbonated calcium hemicarboaluminate $Al\,Ca_2C_{0.4}O_{9.2}$ | 3.2 (1) | 61 (5) |

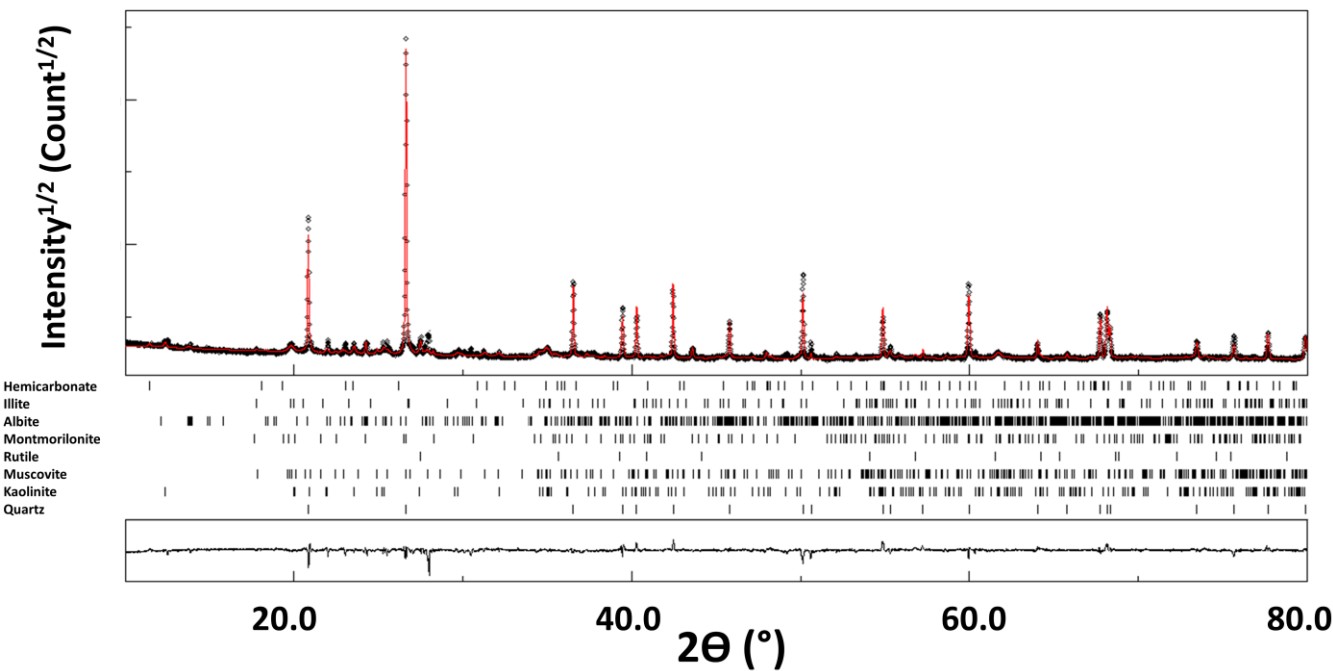

**Figure 11.** XRD pattern of soil 1. The computed pattern (red line) is superimposed on the experimental pattern (black dots). In the center, the fitted phases, and at the bottom, the difference curve (Iobs−Icalc).

### 3.2. Clegg Impact Value as Function of Water Content

Results from the Clegg Impact Soil Tester are reported in Table 3. Thus, the evolution of the Clegg Impact Value as a function of water content is represented in Figure 12. On this latter, it can be seen that CIV increases with the decrease in cob's water content. For water content ranging between 19 wt.% and 0 wt.%, Clegg Impact Value varies from 4 to 35 following a quasi-exponential law. As announced in the introduction, CIV measurements will allow us to propose a field test, allowing us to deduce the in situ cob's compressive strength and water content and in-fine to know if a lift is sufficiently dry and resistant to receive a new lift on its upper side.

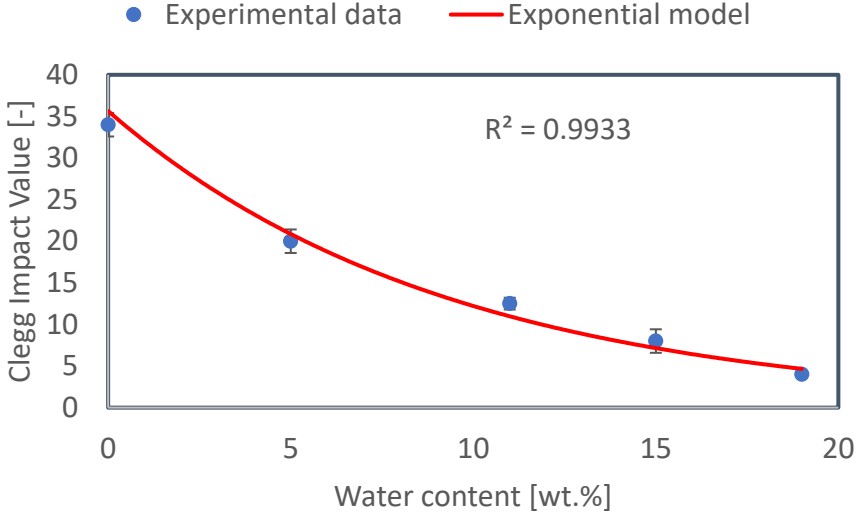

**Figure 12.** Evolution of Clegg Impact Value as function of cob's water content.

### 3.3. In Situ Water Content and Compressive Strength Evolution

Figure 13 shows the course of the volumetric water content and associated compressive strength (at 1% deformation). As observed in the figure, cob's volumetric water content has decreased gradually with time and tends to a practical value after several months. Moreover, it can be shown that the drying speed becomes slower with the decrease in the water content in the wall. The drying presents two phases: a first one that is faster and a second one slower. With water content getting lower, the drying process gets slower. This drying is consistent with the typical behavior of construction materials.

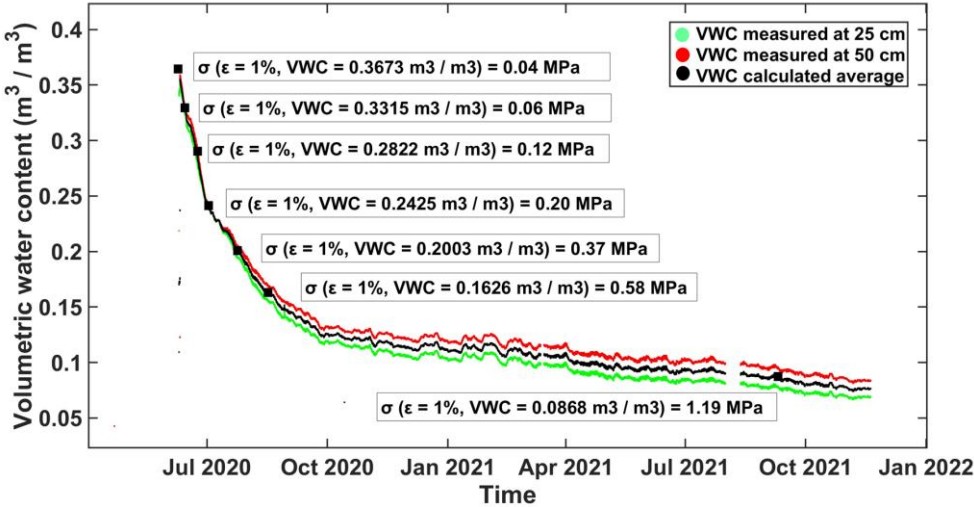

**Figure 13.** Evolution of cob's compressive strength with the water content course.

With the decrease in water content, the cob's compressive strength increases. In this figure, it can be observed that compressive strength passes from 0.04 to 0.06, 0.12, 0.20, 0.37, and 0.58 MPa after 04, 15, 22, 44, and 68 days, respectively. After approximately one year and two months, compressive strength is approaching 1.2 MPa.

In Figure 13, the reported water content is recovered in situ, but mechanical performances are obtained on samples produced in the laboratory and tested at determined water contents considering those encountered in situ.

In the studied prototype, formworks are released after 27 days, and a new lift was raised directly. The choice of this delay is based on Norman craftsmen practice. At that time, the VWC of the existing lift was approx. $0.2278 \text{ m}^3/\text{m}^3$ corresponding to a compressive strength of approx. 0.28 MPa. From this experience, it can be affirmed that 0.28 MPa (reached after approx. 27 days in this typical case and weather conditions) is sufficient to raise up a new lift. However, this value can be optimized by performing judicious calculations.

In this sense, calculations were undertaken by considering the stress imposed by the new wet lift (layer of wet cob) on the existing lift. These calculations are based on the knowledge of the wet cob density, its volume, gravitational acceleration, and existing lift horizontal surface (see Equation (2)).

$$\sigma = \frac{N}{S} = \frac{m \cdot g}{S} = \frac{\rho \cdot V \cdot g}{S} \qquad (2)$$

where N is the normal stress, m is the wet cob mass, ρ is the wet cob density, V is the wet cob volume, g is the gravitational acceleration, and S is the surface of the existing lift.

Considering the measured density of the wet cob, it was found that a new wet cob lift (height = 70 cm) exerts a stress of 0.013 MPa directly after its implementation. This value seems to be too low compared to that of compressive strength measured at $0.3673 \text{ m}^3/\text{m}^3$ (water content of the newly implemented wet cob), which is equal to 0.04 MPa (for a

deformation of 1%). This would imply that a cob lift could support a new wet lift from the first day after its implementation. But, this seems to be too optimistic. Consequently, when trying to calculate the minimum mechanical resistance that a cob lift should reach before raising up a new wet lift, other methods than the one presented here should be identified and considered.

*3.4. Clegg Impact Value and Compressive Strength*

In the previous subsections, it was seen that both cob's compressive strength and CIV increase exponentially with the decrease in water content. Thus, the evolution of Clegg Impact Value is plotted as a function of compressive strength. Figure 14 shows that the evolution of CIV as a function of σ is best fitted via a linear relationship. On this latter, it can be seen that the value of compressive strength obtained after 27 days of drying (approx. 0.28 MPa) correspond to a CIV value of approx. 9. As it was already reported that this compressive strength is sufficient to raise up a new lift, it can be stated here that in situ measurement of a CIV value equal to or larger than 9 is sufficient for raising up a new lift on the existing one. More on-site investigations could help to optimize this CIV value. However, when dealing with soils different from those reported in this present study, the absolute values of compressive strength and Clegg Impact Value should be considered carefully.

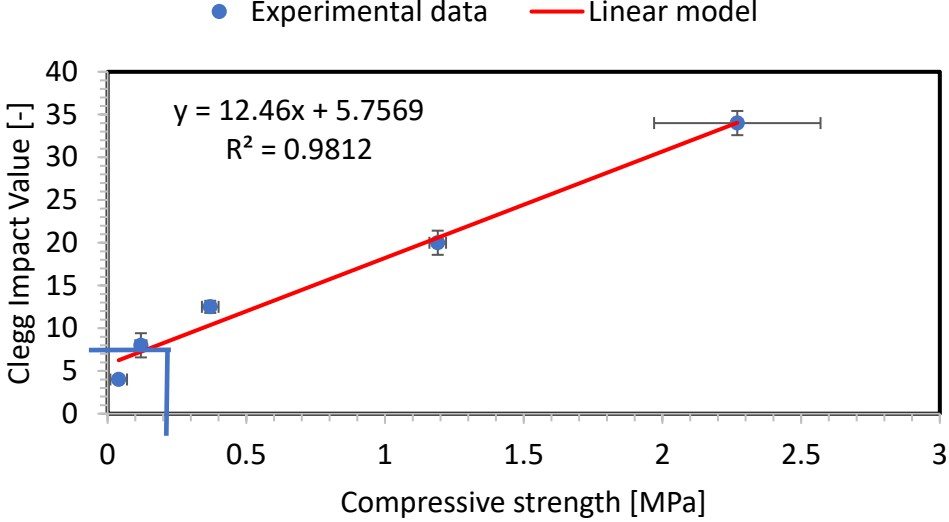

**Figure 14.** Relationship between cob's compressive strength (at 1% deformation) and Clegg Impact Value.

## 4. Conclusions

The first aim of this present work was to understand the link between cob's drying process and its mechanical performances in a CobBauge double-walling system. This is consistent with the objective of better controlling construction times. In this regard, laboratory and in situ studies were performed. First, the water content was recovered during the construction of a cob wall (under real conditions). These measurements showed that water content decreases exponentially with time. Then, samples were produced in the laboratory at the water contents encountered in situ. Afterward, these samples were subjected to compressive strength tests at different water contents. When considering the decrease in cob's water content, as shown in the in situ measurement, compressive strength was found to follow an exponential increase.

The second aim of this present work was to propose a simple tool allowing evaluation of cob's hydromechanical performances in situ. Thus, the Clegg Impact Soil Tester (CIST) was proposed. Indeed, this can represent an easy way for craftsmen to control in situ cob drying and readiness to receive a new lift. In this sense, samples were also subjected to the CIST in order to obtain the Clegg Impact Value (CIV) at different water contents. The

obtained results showed that the evolution of CIV as a function of water content can be best fitted via an exponential model.

From there, the evolution of Clegg Impact Value is plotted as a function of compressive strength (σ). It was found that the evolution of CIV as a function of σ is best fitted via a linear relationship.

In the studied wall, a new lift was raised up on the monitored one after 27 days. In this present study, it was found that after this duration, volumetric water content (VWC) in the wall was approximately equal to 0.2278 $m^3/m^3$. At this VWC, the cob presents a compressive strength of approx. 0.28 MPa. When CIV was plotted as a function of compressive strength, it was found that 0.28 MPa corresponds to a CIV value of approximately 9. As the form striping and the raise up of a new lift at this compressive strength went well, it can be stated that measuring a CIV value of at least 9 can allow the rise of a new CobBauge lift safely.

**Author Contributions:** Conceptualization, K.T., M.L.G., F.S. and S.G.; methodology, K.T., B.A.S. and M.L.G.; investigation, data curation, and formal analysis, K.T., B.A.S., M.L.G. and Y.E.M.; writing—original draft preparation, K.T., B.A.S., M.L.G. and Y.E.M.; writing—review and editing, K.T., Y.E.M., F.S. and S.G.; funding acquisition, S.G. and F.S. All authors have read and agreed to the published version of the manuscript.

**Funding:** The results presented in this article were obtained in the framework of the collaborative project CobBauge, funded by the European cross-border cooperation program INTERREG V France (Manche/Channel) England.

**Informed Consent Statement:** Not applicable.

**Data Availability Statement:** The experimental and computational data presented in this present paper are available from the corresponding author upon request.

**Conflicts of Interest:** The authors declare that they have no known competing financial interest or personal relationship that could have appeared to influence the work reported in this paper.

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
