# Peer review of "Insight into the Optimization of Implementation Time in Cob Construction: Field Test and Compressive Strength Versus Drying Kinetics"

_2673-4117, doi:10.3390/eng4030117_

Round 1
Reviewer 1 Report
In their paper entitled “Insight into the optimization of implementation time in cob construction: field test and compressive strength versus drying kinetics” authors propose an adaptation of the Clegg Impact Value (CIV) in order to assess the bearing capacities of cob lifts during drying. This subject is of great interest, since the current on-site evaluation is made manually. The use of CIV for cob building is novel and promising. It would provide a rational tool to ease the engineering control and speed up cob construction. Nonetheless, some points needs to be addressed in order to strengthen the paper. Authors are encouraged to take into consideration the following comments.
General comment: some mistakes are present in the paper, please, check the English and correct these errors
l. 102: you write that 3 soils were used for construction and only 2 soils are presented in Table1, which are may be different from the 3 soils used for construction. This is confusing. Can you precise this point?
Table 1: Regarding the MBV the PI should be in the region of 10 to 15 %, PI of Soils 1 and 2 are therefore very low, could you check these values?
l. 120: “It is composed of minerals that have not been altered, such as quartz grains and silicates (feldspars, micas, serpentines, and smectites)”, this is just a detail, but smectite and serpentines are clearly produced by the alteration of rocks, so this assertion might be tempered.
l. 133-136: which soil has been used to produce the specimens? How many repetitions were made for each water content? A table summarize the specimen production would help the reader to understand the testing campaign.
l. 139-140: “the mixture is compacted in several layers with a 139 wooden tamper”, how do you ensure a same compaction energy? Have you defined a number of blows? Is it always the same operator?
Figure 2: a detail, the slenderness ratio of the specimens on the picture seems closer to 3 than to 2. May be the picture has been resized without conserving the proportions. This can be confusing for the reader. Please check this.
l. 148-150: the part regarding the CIST test should be moved to section 2.6.
l. 164: is this the wet or the dry bulk density which is measured?
l. 177: “Compressive strength tests are performed on the prepared cylindrical samples 176
(∅110 mm × H220 mm), in accordance with the NF EN 13286-14 standard”, this standard is not available so that it is impossible to check the procedure. Check the standard reference or precise the testing protocol in the paper
Section 2.6: could you explain how the CIV is calculated?
l. 225-227: Is the paragraph “This section may be divided by subheadings. It should provide a concise and precise description of the experimental results, their interpretation, as well as the experimental conclusions that can be drawn” really necessary?
l. 230: it might be more informative to define the state of the material (solid, plastic, liquid …) rather than to talk about water content. At a same water content, a soil can be at solid state while another one can be at plastic state. The Liquid Index can be an option.
l. 235-236: “A deformation of 1% is considered in present work for considerations regarding the ease of straightening wall surfaces before the application of plasters or renders”: I’m not sure about the meaning of “straightening” in this sentence. If the idea is to define a strain threshold of 1% to avoid higher strain that might damage the plaster, I understand well the interest of this threshold at dry/low water content (0-5 %), but not at higher water content, since plasters are implemented after drying of the wall. Or, if I understood wrong, please clarify this point.
Figure 10: If I understand well, the density is the wet density (mass of earth+water). This type of correlation is quite uncommon and usually there is a correlation between DRY density and mechanical strength. In this case, the dry density has little interest. However, I also wonder if there is an interest to exhibit wet density results, since they are not really analysed in the discussion.
l. 238-241: “soils containing more than 30% clay have very high shrinkage/swelling ratios, which, along with their proclivity to absorb moisture, can result in large fissures in the finished cob product and thus impact its mechanical performance”. As said by the authors later in the same paragraph, there are many different types of clays, which have very different swelling properties. The swelling/shrinkage behaviour of a soil depends on the amount of clay AND on the activity of the clay. These 2 parameter are captured by the Specific Surface Area of the soil which can be estimated with the Cation Exchange Capacity or the Methylene Blue Value. Clay content only is not enough to assess soil suitability.
l. 242-261: Authors provide a long and complex discussion regarding the swelling of clays, but without any results supporting the assertions which are, therefore, more speculations rather than results. It has its place in the results and discussion section, but this very speculative assertions may, in my point of view, weaken the message of the paper.
l. 259: “muscovite, albite, kaolinite, and illite. These clays structures” Muscovite is a phyllosilicate but not a clay, and albite id a feldspar and not a clay. The way the sentence are written is confusing and leave the impression that authors consider muscovite and albite as clay. Please correct this point.
l. 261-266: “The analysis of the cob after drying shows the disappearance of huntite (Mg₃Ca(CO₃)â‚„) combined with a decrease in the montmorillonite ((Na,Ca)0.3(Al,Mg)2Si4O10(OH)2) content in favor of the formation of gaylussite (Na₃Ca(CO₃)2.5.H2O) [26]. The formation of carbonate hydrates leads to the increase of materials’ bulk density, which is beneficial for the enhancement of compressive strength, see Figure 10.” I really don’t understand this paragraph … (1) no results of cob after frying are presented, (2) you mean that, just during the drying process, you suppose a chemical reaction between minerals (huntite + montmorillonite => gaylussite?) in the cob mixture, which is quite unlikely to happened, (3) you support this assertion with the citation of a PhD dealing with hydraulic binder adjuventation of earth material, whereas here it is not the case. If this assertion is true, authors are encouraged to develop this part in order to provide a clear and well supported demonstration (I mean with convincing results).
Figure 12: It is not clear if the mechanical strength indicated are estimated after the water content or if they come from tests carried out during several months. Please precise this point.
l. 295: please, define WC
l. 301-306: “From this experience it can be affirmed that 0.28 MPa (reached after approx. 27 days in this typical case and weather conditions) is sufficient to raise up a new lift.” Who has decided to raise the lift after 27 days? Why after 27 days? Please precise these points. Precise also if the new lift has stand or if it has failed.
l. 307-312: “but this seems not to be sufficient”, please, precise what is not sufficient; “In fact, calculations based on the knowledge of the wet cob density allowed a resistance of 0.013 MPa to support a new lift” this sentence is not clear, please revise it; “But this value seems to be too weak” Please explain why; “Factors other than humid density should be identified and considered when trying to calculate the minimum resistance to reach before raising up a new lift.”, please, give some examples or explain more this part
Section 3.4: the threshold value of 9 proposed by the authors, based on 1 successful lift rising is not convincing. May be if the lift had been raised after 23 days, it had stand and the threshold would have been different. Further investigations are required in order to propose an optimized threshold.
General comment: some mistakes are present in the paper, please, check the English and correct these errors
Author Response
Please find the answers in the attached file

Reviewer 2 Report
The article deals with an important subject related to the monitoring of water content and compressive strength in Cob construction to optimize construction times. It aims at finding an in-situ testing method which can provide information about the evolution of compressive strength values, which have direct consequences on the rapidity of execution of the entire Cob walls, by assessing the Clegg Impact Value.
The article proposes an interesting topic which is studied with a sound methodology.
Nevertheless, some corrections should be adopted to enhance the understandability and readability of the article.
In paragraph 2.3 are not explaining how they attain the different water contents for each sample: are they using a climatic chamber? Are they oven drying? Please specify.
Why the samples for the CIV test are not prepared at all the same water contents as those for the compressive strength test? There are 3 conditions left. If possible, please add these values.
In results and discussion section, from line 297, are the compressive strength values of figure 12 those assessed on the cylindrical samples or are they somehow assessed on the cob wall? This passage is not clear; indeed, the authors added information about compressive strength performance evolution over time, but it is not clear if these values are estimated on the base of water content evolution measured on full scale wall or measured.
On the base of what formworks are removed after 27 days? Please add an explanation even if it comes from building practice.
In the materials and methods section, some English spell check should be made (paragraph. 2.2, the first phrase should be rewritten; paragraph 2.3 please substitute “before..beforehand”).
Author Response

(The authors gave the same response as above.)
